# The Role of Corneal Biomechanics for the Evaluation of Ectasia Patients

**DOI:** 10.3390/ijerph17062113

**Published:** 2020-03-23

**Authors:** Marcella Q. Salomão, Ana Luisa Hofling-Lima, Louise Pellegrino Gomes Esporcatte, Bernardo Lopes, Riccardo Vinciguerra, Paolo Vinciguerra, Jens Bühren, Nelson Sena, Guilherme Simões Luz Hilgert, Renato Ambrósio

**Affiliations:** 1Instituto de Olhos Renato Ambrósio, Rio de Janeiro 20520050, Brazil; marcella@barravisioncenter.com.br (M.Q.S.); louisepgomes@hotmail.com (L.P.G.E.); blopesmed@gmail.com (B.L.); 2Rio de Janeiro Corneal Tomography and Biomechanics Study Group, Rio de Janeiro 20520050, Brazil; 3Brazilian Study Group of Artificial Intelligence and Corneal Analysis—BrAIN, Rio de Janeiro 20520050, Brazil; 4Department of Ophthalmology, Federal University of São Paulo, São Paulo 04023062, Brazil; analhofling@gmail.com; 5Instituto Benjamin Constant, Rio de Janeiro 22290255, Brazil; 6School of Engineering, University of Liverpool, L69 3GH Liverpool, UK; vinciguerra.riccardo@gmail.com; 7Humanitas San Pio X Hospital, 20159 Milan, Italy; 8The Eye Center, Humanitas Clinical and Research Center, 20089 Rozzano, Italy; paolo.vinciguerra@humanitas.it; 9Vincieye Clinic, 20141 Milan, Italy; 10Praxis für Augenheikunde Prof. Bühren, D-60431 Frankfurt, Germany; buehren@ophthalmologicum-ffm.de; 11Department of Ophthalmology, Federal University the state of Rio de Janeiro (UNIRIO), Rio de Janeiro 22290-240, Brazil; nj88@globo.com; 12Hospital da Gamboa, Rio de Janeiro 20220-324, Brazil; guigarcia_@hotmail.com

**Keywords:** keratoconus, corneal ectasia, corneal tomography, corneal biomechanics

## Abstract

Purpose: To review the role of corneal biomechanics for the clinical evaluation of patients with ectatic corneal diseases. Methods: A total of 1295 eyes were included for analysis in this study. The normal healthy group (group N) included one eye randomly selected from 736 patients with healthy corneas, the keratoconus group (group KC) included one eye randomly selected from 321 patients with keratoconus. The 113 nonoperated ectatic eyes from 125 patients with very asymmetric ectasia (group VAE-E), whose fellow eyes presented relatively normal topography (group VAE-NT), were also included. The parameters from corneal tomography and biomechanics were obtained using the Pentacam HR and Corvis ST (Oculus Optikgeräte GmbH, Wetzlar, Germany). The accuracies of the tested variables for distinguishing all cases (KC, VAE-E, and VAE-NT), for detecting clinical ectasia (KC + VAE-E) and for identifying abnormalities among the VAE-NT, were investigated. A comparison was performed considering the areas under the receiver operating characteristic curve (AUC; DeLong’s method). Results: Considering all cases (KC, VAE-E, and VAE-NT), the AUC of the tomographic-biomechanical parameter (TBI) was 0.992, which was statistically higher than all individual parameters (DeLong’s; *p* < 0.05): PRFI- Pentacam Random Forest Index (0.982), BAD-D- Belin -Ambrosio D value (0.959), CBI -corneal biomechanical index (0.91), and IS Abs- Inferior-superior value (0.91). The AUC of the TBI for detecting clinical ectasia (KC + VAE-E) was 0.999, and this was again statistically higher than all parameters (DeLong’s; *p* < 0.05): PRFI (0.996), BAD-D (0.995), CBI (0.949), and IS Abs (0.977). Considering the VAE-NT group, the AUC of the TBI was 0.966, which was also statistically higher than all parameters (DeLong’s; *p* < 0.05): PRFI (0.934), BAD- D (0.834), CBI (0.774), and IS Abs (0.677). Conclusions: Corneal biomechanical data enhances the evaluation of patients with corneal ectasia and meaningfully adds to the multimodal diagnostic armamentarium. The integration of biomechanical data and corneal tomography with artificial intelligence data augments the sensitivity and specificity for screening and enhancing early diagnosis. Besides, corneal biomechanics may be relevant for determining the prognosis and staging the disease.

## 1. Introduction

Understanding the material and structural properties of the cornea has gained increased clinical utility over the latest decades. Knowledge of fundamental biomechanical principles has been applied in several clinical conditions [1,2], including correcting intraocular pressure measurements [3,4], planning and following corneal collagen crosslinking treatments [5,6,7,8], and elective keratorefractive surgery [9,10]. Specifically, in the field of refractive surgery, the investigation of corneal biomechanical properties turned out to be a very relevant part of the screening process, in an attempt to identify patients at higher risk (susceptible) to develop iatrogenic ectasia after laser vision correction (LVC) [9,10,11]. Due to unsatisfactory quality of vision with glasses and contact lenses, patients with keratoconus (KC) and other ectatic disorders, particularly with subclinical disease, frequently present as refractive surgery candidates. These cases typically have from less optimal to poor results after LVC and are at very high risk for iatrogenic ectasia development after the surgical procedure [12,13]. Furthermore, the advent of collagen crosslinking and other treatment modalities such as intracorneal ring segments (ICRS) has made relevant the identification of milder or subclinical forms of ectatic corneal diseases along with monitoring the disease progression [14,15].

The introduction of Placido disc-based corneal topography increased our ability to identify ectatic corneal diseases in earlier stages, before the development of clinical signs or visual symptoms [16,17]. The evaluation of the corneal surface evolved into 3D corneal tomography, with the reconstruction of front and back corneal surfaces along with a full-thickness map [18]. Studies have demonstrated higher sensitivity of the tomographic approach to detect subclinical (or fruste) disease in eyes with “innocent” topographic maps from patients with the fellow eye presenting with clinical ectatic disease [19,20]. Further improvement in corneal shape investigation is related to the ability for segmental or layered tomography with epithelial thickness and Bowman’s curvature mapping [21]. Interestingly, the current concept is that the pathophysiology of ectatic diseases is associated with a primary biomechanical abnormality, with architecture and morphology instability being secondary events [22,23]. Thereby, even with such developments in corneal shape analysis, biomechanical assessment is necessary to enhance the ability to characterize ectasia susceptibility [9]. The purpose of this article is to provide a perspective literature review of the role of corneal biomechanics in the screening, diagnosis, staging, prognosis, and treatment planning for patients with corneal ectasia. We discuss the latest developments in biomechanics investigation and describe the developments, calculation, and performance of new indices from the commercially available instruments, through the analysis of a published cohort of cases, along with anecdotal clinical examples.

### 1.1. THE ORA (Ocular Response Analyzer)

In vivo corneal biomechanical assessment became available with the introduction of the Ocular Response Analyzer (ORA; Reichert, Buffalo, NY, USA) in 2005 by David Luce [24]. This noncontact tonometer (NCT) was designed to provide measurements of intraocular pressure after understanding and compensating for biomechanical properties [24]. Once the measurement starts, an air jet is applied to the eye, which causes the cornea to move inward and to pass through an applanation moment, up to a concavity configuration. As the applied force decreases, the cornea begins to return to its natural contour, passing again through a second applanation time. The pressure is measured and registered at both applanation times. Corneal deformation is monitored by an electro-optical system involving a collimated beam of Infrared Light (IR) light and a photodetector. The ORA provides two main corneal biomechanical parameters: corneal hysteresis (CH) and corneal resistance factor (CRF). Although CH and CRF present lower values in KC eyes as compared to healthy eyes [25], studies have demonstrated a considerable overlap in the distributions of these metrics, so that sensitivity and specificity for proper diagnostic accuracy can be considered somewhat poor [26,27]. Superior results were found with the analysis of the ORA waveform signal and the development of new waveform-derived variables [28,29,30]; and also after the integration of these new parameters with tomographical data [31].

### 1.2. Brillouin Spectroscopy

Another approach newly employed in the investigation of corneal biomechanical properties is Brillouin spectroscopy. The principle of Brillouin light scattering is based on the interaction of light and the intrinsic acoustic waves within the tissue. The acoustic waves are related to molecules’ vibration, naturally present in the tissue at room temperature. When the laser interacts with the tissue, it experiences a Doppler frequency shift, which is proportional to the speed of sound and associated with the elastic modulus [32]. Thereby, this technology allows for the biomechanical characterization of the cornea, crystalline lens, and sclera [33,34,35,36]. 

The application of Brillouin spectroscopy for the diagnosis of ectatic corneal diseases has been investigated. It was found that ex vivo ectatic corneas have a highly significantly smaller Brillouin frequency shift than normal corneas [37]. Further investigators proposed the use of this technology to identify focal weakening in the elastic modulus and found significant differences between Brillouin measurements in the cone region versus in other corneal loci in vivo [32]. Considering the “focal” nature of KC and ectatic diseases, this biomechanical corneal “mapping” might be relevant for allowing earlier detection of the disease. Interestingly, mild KC was discriminated from normal and clinically ectatic corneas comparing the Brillouin frequency between eyes, with biomechanical asymmetry being an efficient and accurate metric [38,39].

### 1.3. Corneal Visualization with Scheimpflug Technology: The Corvis ST 

The Corvis ST is a noncontact tonometer with an ultra-high-speed Scheimpflug camera that acquires 4300 frames per second, allowing for dynamic observation of corneal deformation response [40]. Similarly to what happens with the ORA, as an air puff is triggered, the cornea deforms inwards up to a first applanation and then into a concave shape. While the air puff decreases, the cornea recovers in the outward direction and undergoes a second applanation before returning to its natural position. Once the measurement is executed, the device provides a set of corneal deformation parameters, based on the dynamic inspection of the corneal response (Table 1).

The relevance and application of dynamic Scheimpflug imaging for several clinical conditions were demonstrated in a film produced by Ramos and collaborators, available at https://www.youtube.com/watch?v=VQj1pVexW8c. These include the calculation of accurate measurements of intra-ocular pressure (IOP) [41,42], the investigation of collagen crosslinking results [43], and screening refractive surgery candidates [44], among others. Valbon and coworkers described the correlation between age and biomechanical behavior in healthy eyes and found that the highest concavity time was significantly correlated with age [45]. Faria-Correia et al. evidenced the role of dynamic corneal deformation response for correct IOP measurements in a case of ocular hypertension in pressure-induced stromal keratopathy, which had been misdiagnosed as diffuse lamellar keratitis after LASIK [46].

The role of the Corvis ST for ectasia recognition has been investigated since its prototype. However [9], the first original set of corneal deformation parameters had a relatively poor performance in distinguishing healthy and KC eyes [47,48]. While some improvements have been accomplished with the help of artificial intelligence [44], more recently, superior results have been found following the development and introduction of new deformation parameters [49,50,51,52,53].

In 2014, the first international research task force was created, and the primary purpose of this group was to improve knowledge about the Corvis ST technology, with a particular focus on the investigation of ectatic disease. We describe two main Corvis ST parameters developed by this group for enhancing ectasia detection, the CBI, and the tomographic-biomechanical parameter (TBI). 

### 1.4. The Corneal/corvis Biomechanical Index—CBI

In a retrospective study, one eye randomly selected from 227 healthy and 102 keratoconus patients from the Rio de Janeiro Corneal Tomography and Biomechanics Study Group, Rio de Janeiro, Brazil (Database 1) and from 251 healthy and 78 keratoconus patients from the Vincieye Clinic in Milan, Italy (Database 2) were enrolled. Sixteen different dynamic corneal response parameters were evaluated, including the speed of corneal apex at first applanation (A1 velocity), the speed of corneal apex at second applanation (A2 velocity), the distance between the two bending peaks on the cornea at the maximum concavity state (peak distance), the radius of the central cornea at the maximum concavity state, based on a parabolic fit (highest concavity radius), the deformation amplitude (the most significant displacement of corneal apex in the anterior posterior direction at the moment of highest concavity), deflection amplitude (displacement of corneal apex in reference to the cornea’s initial state), deflection area (“displaced” area of the cornea in the analyzed horizontal sectional plane), inverse concave radius, central –peripheral deformation amplitude ratio, deflection amplitude ratio, Delta Arclength, Ambrósio’s Relational Thickness to the horizontal profile (ARTh), SP – A1 (resultant pressure divided by deflection amplitude at A1), bIOP, corneal thickness, SD- deformation amplitude, among others. 

The best combination of each indices was determined by logistic regression analysis for the development of the corneal biomechanical index (CBI). These parameters included: bIOP, pachymetry (central corneal thickness), deformation amplitude, SD–deformation amplitude, applanation 1 velocity, peak distance, HCdArclength, highest concavity deflection area, deformation amplitude ratio 2 mm, deformation amplitude ratio 1 mm, deflection amplitude ratio, inverse concave radius, radius highest concavity, stiffness parameter-A1, and ARTh. The training dataset was calculated using Database 1, and the best cut-off point was obtained from receiver operating characteristic (ROC) curves, for an accurate separation between healthy and keratoconic eyes. Subsequently, in order to avoid overfitting and independently validate the developed parameter, CBI was tested in a validation dataset (Database 2). Using a cut-off of 0.5, 98.2% of the cases were correctly classified in the training dataset (Dataset 1), with 100% specificity and 94.1% sensitivity and an AUC of 0.983. Then, in the validation dataset (Dataset 2), the same cut-off value correctly classified 98.8% of cases, with 98.4% specificity and 100% sensitivity and an AUC of 0.999 [54]. More recently, a novel algorithm has applied artificial intelligence to develop a new CBI index to discriminate ectatic eyes from stable post-LVC cases.

### 1.5. The Tomographic Biomechanical Index—TBI

The second retrospective study involving patients from Instituto de Olhos Renato Ambrósio in Rio de Janeiro, Brazil and the Vincieye Clinic in Milan, Italy aimed to develop a combined index for the Pentacam and Covis ST. Besides the normal eyes (group 1) and KC eyes (group 2), this study also involved unoperated ectatic eyes from patients with very asymmetric ectasia (group 3) who presented fellow eyes with normal topographic maps (group 4). For groups 1 and 2, only one eye per patient was selected randomly for the inclusion in the study, in order to avoid the bias of the relation between eyes. Data from Pentacam HR and from Corvis ST were exported to a custom spreadsheet using special research software. Different artificial intelligence methods were tested, including logistic regression analysis with forwarding stepwise inclusion, support vector machine, and random forest, which were applied to analyze and combine data from corneal deformation response, including CBI, with tomographic data, including BAD-D; to optimize our ability to separate normal and altered eyes. The best cut-off point was obtained from receiver operating characteristic (ROC) curves for an accurate separation between healthy and keratoconic eyes. With a cut-off value of 0.79, TBI had 100% sensitivity and specificity to detect frank ectasia cases (AUC = 1.0 in groups 2 and 3); however, for the correct characterization of eyes with standard topography having no definitive signs of ectasia from patients with clinical ectatic disease in the fellow eye, and optimization of cut-off value was necessary, and a value of 0.29 provided 90.4% sensitivity with 4% false-positive results (96% specificity; AUC = 0.985). The AUC of the TBI was statistically higher than all other analyzed parameters, including the CBI [55]. Posterior external validation studies were conducted, and the ability of this new index to detect ectatic disease, even in milder forms, has been proven [52,56,57].

## 2. Methods

In the present study, we combined the database of the populations from the original TBI study [55] along with the database of one of the subsequent validation studies [57] to compare the ability of tomographical parameters, biomechanical parameters, and the integration of both approaches for discriminating ectatic disease. A total of 736 normal eyes (group 1), 321 KC eyes (group 2), and 113 unoperated ectatic eyes from patients with very asymmetric ectasia (group 3), who presented fellow eyes (125 eyes) with normal topographic maps (group 4), were included. Tomographic parameters included inferior-superior value (ISV), Belin–Ambrosio Display D value (BAD-D) [19,58,59], Pentacam random forest index (PRFI) [60] and the Pentacam Topographic Keratoconus classification index (TKC). Biomechanical parameters included SP A1, inverse radius, DA Ratio, and CBI. The integrated tomographic-biomechanical parameter (TBI) was also evaluated. The analysis was performed for testing the discriminating abilities to separate normal cases and all ectasia cases, normal cases from the cases with frank ectasia, and normal cases with the supposed subclinical cases. Receiver operating characteristic (ROC) curves were plotted to determine the discriminative ability of each parameter and to obtain critical values (cutoff values) that allow classification with maximum accuracy. Besides the area under the ROC curve (AUROC), separation curves that plot accuracy as a function of a critical value were plotted. The area under the normalized separation curve (A_z_SEP) was calculated between *x* limits of -2 and 2 standard deviations and *y* limits of 50% and 100% accuracy as described before [Ambrósio JRS 2016] using a custom-written MATLAB program (R14, The MathWorks, Natick, Mass). Pairwise comparisons of the AUCs were achieved with a nonparametric approach as described by DeLong and coworkers for comparing the performance of diagnostic tests [61].

## 3. Results

Comparing all groups, the AUC of the TBI to detect ectasia (groups 2, 3 and 4) was found to be 0.992, which was statistically higher than PRFI (0.982), BAD-D (0.959), CBI (0.91), IS Abs (0.91), TKC index (0.858), SP A1 (0.857), DA Ratio (0.84), and Inverse Radius (0.83) (Figure 1). AUSEP- Area under the separation curves revealed a higher spread among the metrics, with the TBI reaching 137 and the IS Abs a value of 45 (Figure 2). The TBI had almost 100% sensitivity to detect frank ectasia (AUC 0.999 and AUSEP 169) in groups 2 and 3 (Figure 3 and Figure 4), being once more statistically higher than all other parameters described, including PRFI (0.996; 116), BAD-D (0.995; 70), CBI (0.949; 102), TKC Index (0.952; 47), and IS Abs (0.977; 57). When evaluating the VAE-NT group, the AUC of the TBI was 0.966, which was statistically higher than PRFI (0.934), BAD- D (0.834), CBI (0.774), IS Abs (0.677), TKC index (0.531), SP A1 (0.712), DA Ratio (0.722), and Inverse Radius (0.670) (Figure 5). Here also, a high spread of AUSEP values with the TBI reaching a score almost twice as high as the PRFI (113 vs. 66) could be observed (Figure 6). Thus, TBI had a statistically higher AUC (DeLong, *p* < 0.001) than all other parameters tested for every analysis performed (Figure 7). 

## 4. Conclusions

In all groups tested, the tomographic biomechanical index (TBI) presented a better performance for diagnosing corneal ectatic disease compared to all other investigated parameters, including PRFI, BAD-D, CBI, IS- Abs, TKC index, SP A1, DA Ratio, and Inverse Radius. The TBI had almost 100% sensitivity to detect frank ectasia in groups 2 and 3. This finding is in agreement with what we expected, as these patients already presented clinical ectasia, and biomechanical abnormalities should be expected. Interestingly, when evaluating the VAE-NT group, it was found that the AUC of the TBI was statistically higher than all other parameters described as well, and this finding suggests that the TBI increased the ability to detect early ectasia, even in eyes with no clinical ectasia. 

This article demonstrates that the integration of biomechanical characterization with tomography augments accuracy for detecting ectatic disease and also identifying the susceptibility for ectasia progression. New algorithms using artificial intelligence are possible and are under development for this purpose. Further integration with other multimodal imaging tools (i.e., segmental tomography of the cornea, ocular wavefront) may result in higher accuracy. 

Furthermore, the biomechanical data has demonstrated to be relevant not only for the correct diagnosis but also for a proper follow-up post-LVC. Prognostic information may also be used, which may be critical for clinical decisions in ectatic cases. 

In conclusion, corneal biomechanics may be relevant for diagnosis, for determining the prognosis, staging the disease, and also for customizing the treatment planning. 

The clinical applicability of corneal biomechanical investigation is demonstrated below with anecdotal clinical examples. 

### 4.1. Clinical Example 1

A 17-year-old teenager was referred for a second opinion related to a diagnosis of unilateral keratoconus. This patient complained of progressive visual acuity decreased in the right eye but was concerned about preserving his vision in the left eye. Figure 8 demonstrates the TBI display from the Pentacam and Corvis integration. We can note that front surface curvature analysis showed an advanced KC on the right eye (Figure 8A—upper right part of the display) and a relatively normal topographic map on the left eye (Figure 8B—upper right part of the display). Deformation corneal response was investigated in both eyes, and interestingly, despite a typical topographic map, the tomographic biomechanical display (TBI) from Pentacam and Corvis ST revealed an abnormally high TBI value of 0.70 (cut-off 0.25 to 0.30) in the left eye, even though CBI (0.00) and BAD-D (1.05) presented normal values (Figure 8B). Treatment was proposed, the patient was concerned with the bilaterality and asymmetry of the ectatic disease, and he was warned not to rub both eyes and informed of the need to properly follow up with corneal imaging.

This case is exciting as it raises the hypothesis that a subclinical condition could actually occur in both eyes of the same patient [62], as in the identical twin case report from Guerra and coworkers [63], in which one sister had clinical keratoconus in one eye and a subclinical condition in the fellow eye, while her twin had a bilateral subclinical presentation. On the other hand, corneal ectasia might occur unilaterally, as described by Ramos and coworkers, who presented a case of unilateral ectasia, in which the fellow eye presented with a healthy cornea, which was documented stable over four years later with advanced diagnostic methods including corneal topography, tomography, segmental tomography, and biomechanical assessment [64]. 

### 4.2. Clinical Example 2

This 22-year-old male patient came for his first consultation on 27/03/2017 with a recent diagnosis of keratoconus. Slit-lamp biomicroscopy showed more evidenced corneal nerves and Voght striae in both eyes, especially in OD. Corneal topography revealed an advanced KC in OD, with a K Max value of 71.0 D, and moderate KC in OS, with a Kmax value of 51.3D (Figure 9C,D). Treatment options were discussed and proposed, and the patient was adverted not to rub his eyes and to return for frequent consultations every four months. However, this patient did not return for evaluation until 2019, when we had the chance to perform tomographical and biomechanical analysis once more. Figure 9A,B demonstrates the curvature maps from OD and OS in 2019. We can note a significant progression in the anterior topographic maps from both eyes. With the Corvis ST, the time changes in corneal biomechanical properties could also be investigated in this case. Figure 10 demonstrates the biomechanical comparison display from the right eye. We can note that the cornea becomes softer and thinner, considering the first and second consultations, as noted comparing the variables DA ratio, integrated radius, SP A1, and ARTh. This case demonstrates the role of corneal deformation response analysis not only for diagnosis but also for the prognosis of the disease. 

### 4.3. Clinical Example 3

A 50-year-old male patient complained of a progressive decrease in visual acuity in the latest years and sought refractive surgery in both eyes. Slit-lamp biomicroscopy confirmed that he had undergone hexagonal incision surgery, which was performed thirty years ago for hyperopic and astigmatic correction. Intraocular pressure was 16 mmHg in both eyes, and the fundoscopic exam was unremarkable. Slit-lamp biomicroscopy showed hexagonal corneal incisions in both eyes (Figure 11). Placido disk-based corneal topography evidenced post-surgery corneal ectasia in both eyes, as noted in Figure 12. Interestingly, the Vinciguerra Screening Report from Corvis ST evidenced abnormal biomechanical parameters, even after adjusting the software for post-LVC state. Note the abonormal CBI value for the right eye in Figure 13.

## 5. Discussion

In vivo characterization of corneal biomechanics is a real tool for clinical assessment. Understanding the corneal biomechanical behavior is useful in several clinical conditions in different fields of ophthalmology. Specifically, in the field of refractive surgery, the investigation of corneal biomechanical properties is fundamental to identify patients at higher risk to develop iatrogenic ectasia after laser vision correction and to enhance clinical outcomes. Further prospective studies, primarily involving populations from different countries, are still needed in this promising biomechanical field. 

## Figures and Tables

**Figure 1 ijerph-17-02113-f001:**
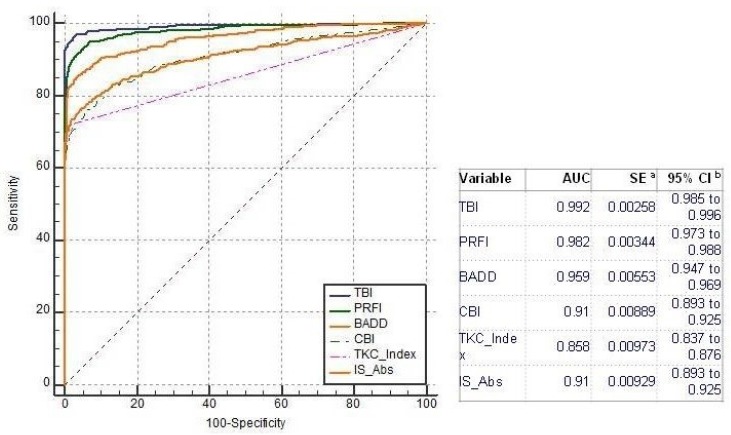
Receiver operating characteristic (ROC) curves from tomographic-biomechanical parameter (TBI), pentacam random forest index (PRFI), Belin–Ambrosio Display D value (BAD-D), corneal biomechanical index (CBI), Pentacam Topographic Keratoconus classification index (TKC) and IS Abs considering groups 2, 3, and 4. Note that the area under the curve AUC of the TBI was statistically higher than all other parameters.

**Figure 2 ijerph-17-02113-f002:**
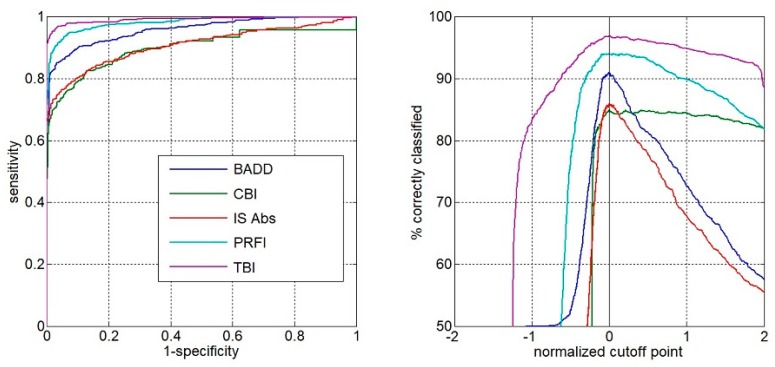
AUSEP- area under the superation curves from TBI, PRFI, BAD-D, CBI, and IS-Abs considering groups 2, 3, and 4. Note the higher spread among the metrics with the TBI.

**Figure 3 ijerph-17-02113-f003:**
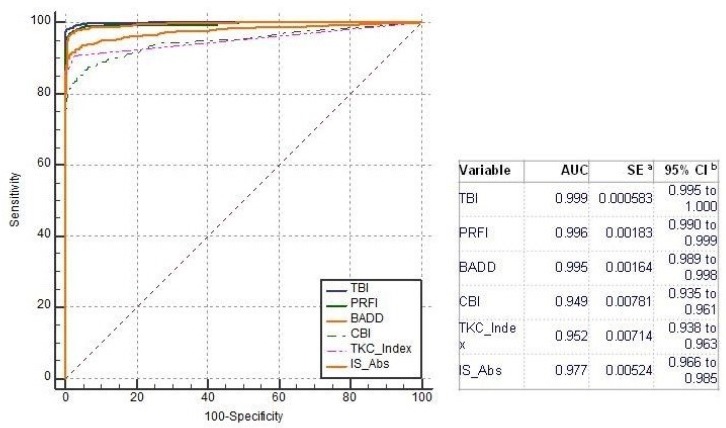
ROC curves from TBI, PRFI, BAD-D, CBI, TKC, and IS Abs considering only groups 2 and 3. Note that TBI had almost 100% sensitivity to detect frank ectasia (AUC 0.999), being once more statistically higher than all other parameters described.

**Figure 4 ijerph-17-02113-f004:**
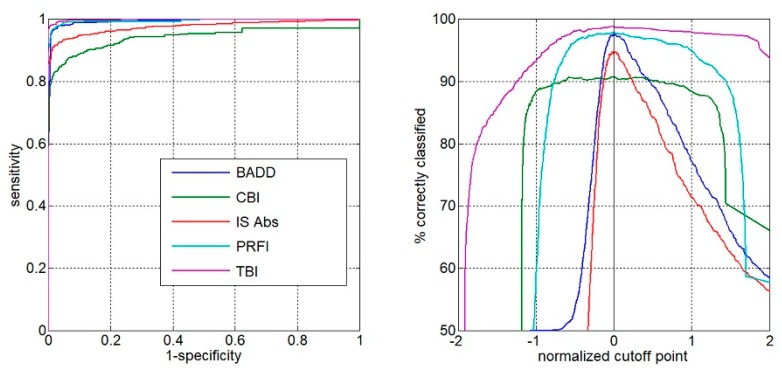
AUSEP curves from TBI, PRFI, BAD-D, CBI, and IS-Abs considering only groups 2 and 3. Note, once more, the higher spread among the metrics with the TBI.

**Figure 5 ijerph-17-02113-f005:**
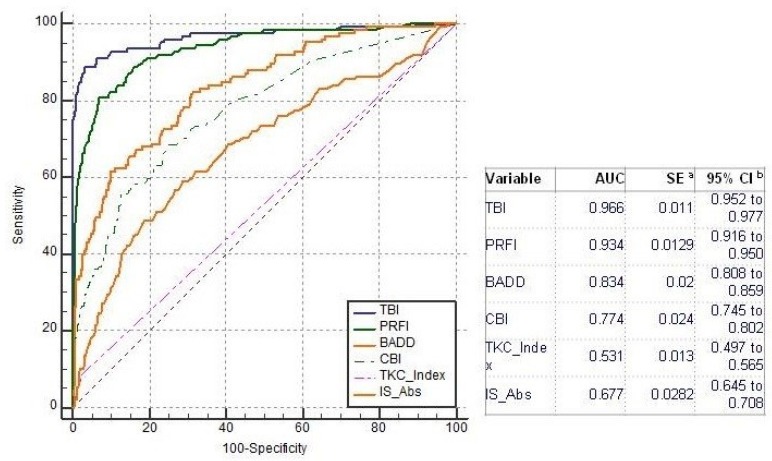
ROC curves from TBI, PRFI, BAD-D, CBI, TKC, and IS Abs considering the with very asymmetric ectasia with relatively normal topography (VAE-NT) group. Note that the AUC of the TBI was 0.966, which was once more statistically higher than all other parameters tested.

**Figure 6 ijerph-17-02113-f006:**
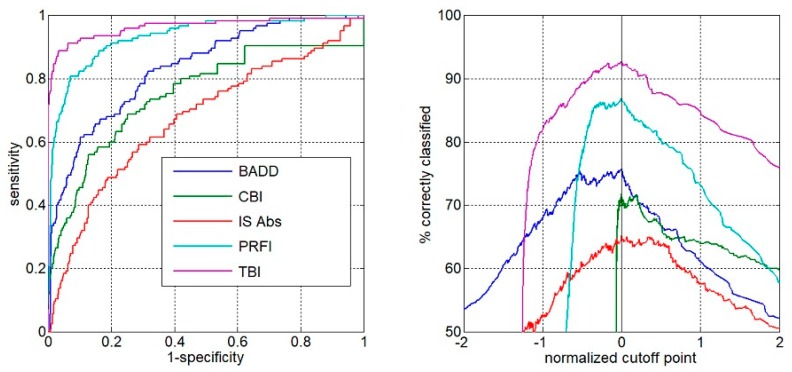
AUSEP curves from TBI, PRFI, BAD-D, CBI, and IS-Abs considering only groups VAE-NT group. Note, once more, the higher spread among the metrics with the TBI.

**Figure 7 ijerph-17-02113-f007:**
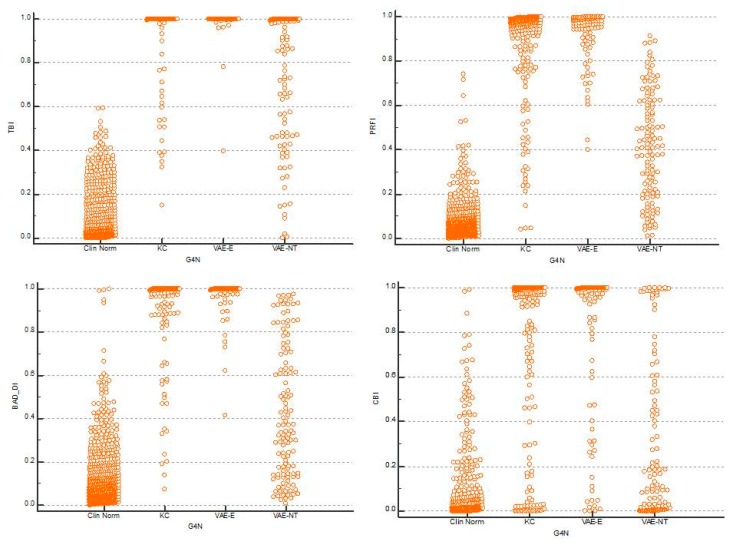
Graphical representation of the data distribution-Dot plots. Note one more time the better performance of TBI compared to PRFI, BAD-D, and CBI.

**Figure 8 ijerph-17-02113-f008:**
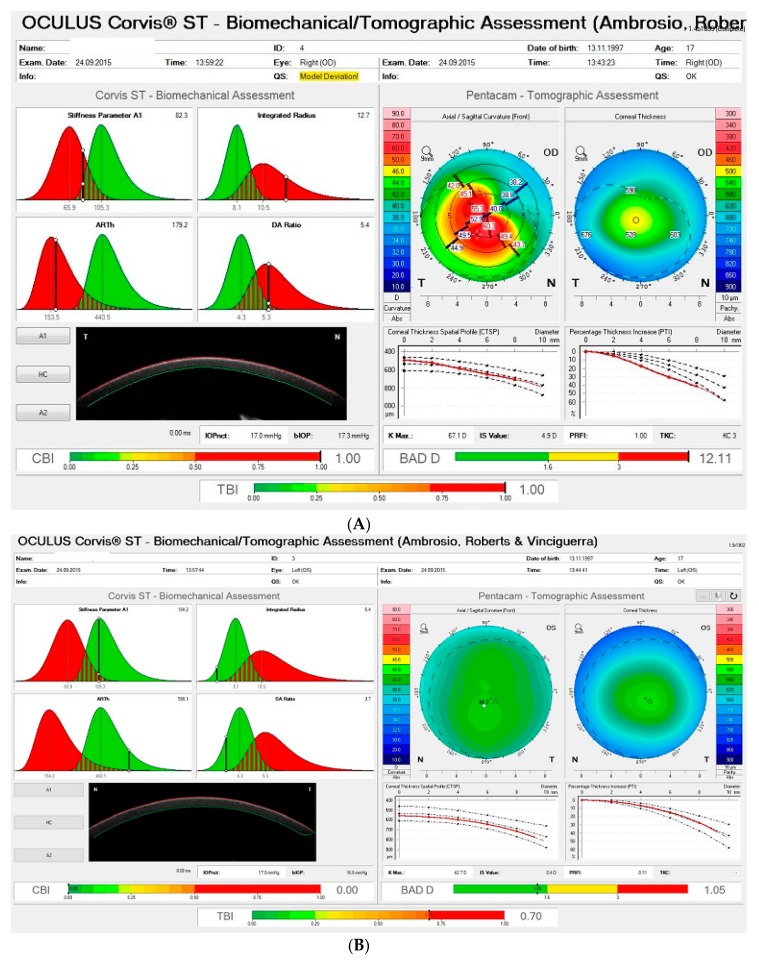
(**A**) Tomographical biomechanical display (ARV) from the right eye. Note, on the Pentacam Tomographic Assessment (top right), that the front surface curvature demonstrates an advanced KC condition on this eye. (**B**) Tomographical biomechanical display (ARV) from the left eye. Note a relatively normal anterior curvature map on the Pentacam Tomographic Assessment (top right). Interestingly, deformation corneal response revealed an abnormal TBI value of 0.70 in the left eye, even though CBI (0.00) and BAD-D presented normal values.

**Figure 9 ijerph-17-02113-f009:**
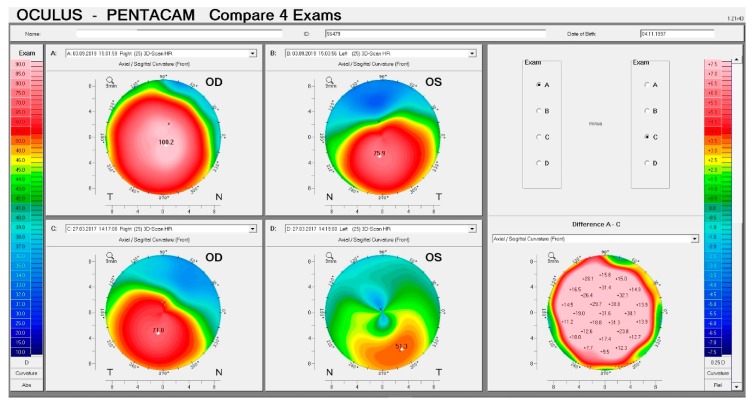
(**A**) and (**C**): Anterior curvature maps from OD in 2019 and 2017, respectively. (**B**) and (**D**): Anterior curvature maps from OS in 2019 and 2017, respectively. The right-hand panel demonstrates the differential maps from OD, which was the eye that presented the highest progression.

**Figure 10 ijerph-17-02113-f010:**
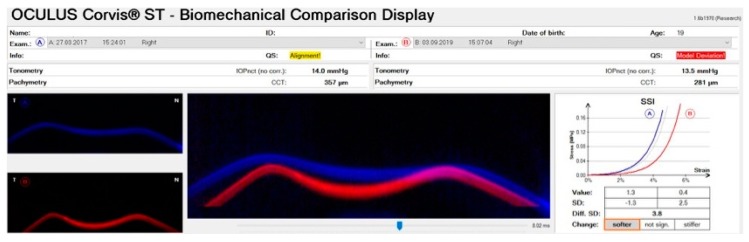
Biomechanical comparison displayed from the right eye in 2017 (Exam A) and 2019 (Exam B). Note that the cornea becomes softer and thinner, considering the first and second consultations, as noted comparing the variables DA ratio, integrated radius, SP A1, and ART h.

**Figure 11 ijerph-17-02113-f011:**
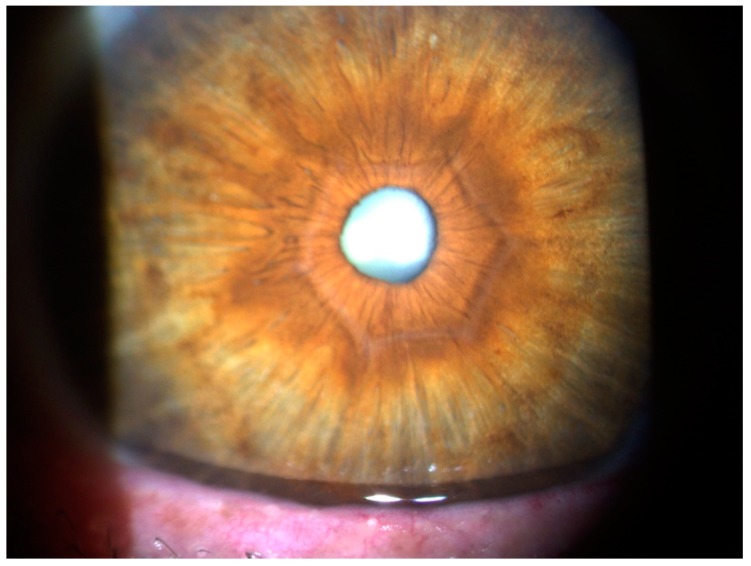
Slit-lamp biomicroscopy photo from the right eye. Note the hexagonal corneal incisions.

**Figure 12 ijerph-17-02113-f012:**
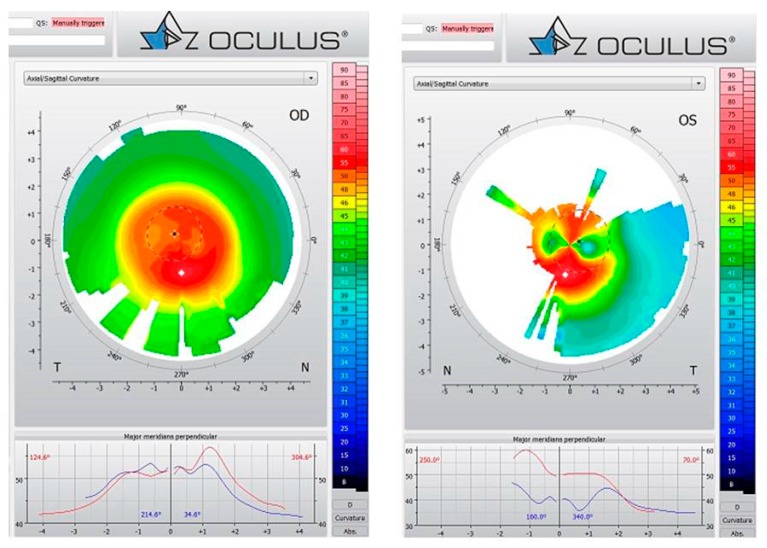
Placido disk-based corneal topography from OD and OS demonstrating the ectatic condition in OU.

**Figure 13 ijerph-17-02113-f013:**
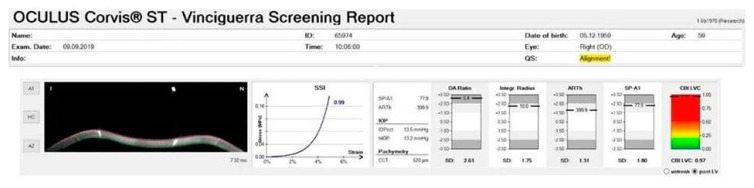
Vinciguerra Screening Report from the right eye evidencing abnormal biomechanical parameters, even after adjusting the software for post-laser-vision-correction (LVC) state. Note the abonormal CBI value.

**Table 1 ijerph-17-02113-t001:** Corneal deformation parameters provided by the Corvis ST.

**1st Applanation**
Moment of first applanation of the cornea during the air puff (in milliseconds). In parenthesis is the length of the applanation at this moment (in millimeters).
**Highest Concavity**
The instant that the cornea assumes its maximum concavity during the air puff (in milliseconds). In parenthesis is the length of the distance between the two peaks of the cornea at this moment (in millimeters).
**2nd Applanation**
The second applanation of the cornea during the air puff (in milliseconds). In parenthesis is the length of the applanation at this moment (in millimeters).
**Maximum Deformation**
Measurement (in millimeters) of the maximum cornea deformation during the air puff.
**Wing Distance**
The length of the distance between the two peaks of the cornea at this moment (in millimeters).
**Maximum Velocity (in)**
Maximum velocity during the ingoing phase (in meters per seconds (m/s)).
**Maximum Velocity (out)**
The maximum velocity during the outgoing phase (in meters per seconds (m/s)).
**Curvature Radius Normal**
Radius curvature of the cornea in its natural state (in millimeters).
**Curvature Radius HC**
Radius of curvature of the cornea at the time of maximum concavity during the air puff (in millimeters).
**Cornea Thickness**
Measurement of the corneal thickness (in millimeters).
**Integrated Inverse Radius**
Inverse of the radius of curvature during the concave phase of the deformation.
**Deformation Amplitude Ratio 1 or 2 mm**
The central cornea deformation divided by an average of the deformation 1 or 2 mm at either side of the center with maximum value just before 1st applanation
**IOP** Measurement of the intraocular pressure (in millimeters of mercury (mmHg)).
**bIOP** Biomechanically-corrected IOP

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
