# Peer review of "The Role of Corneal Biomechanics for the Evaluation of Ectasia Patients"

_ijerph, 2020, doi:10.3390/ijerph17062113_

Round 1
Reviewer 1 Report
As the authors make clear, biomechanical measurements of the cornea may, in principle, give valuable additional information to supplement other methods for the diagnosis, prognosis and monitoring of ectatic conditions. The present manuscript offers a review of available methods for quantifying biomechanical results in terms of various indices and applies these to the analysis of a large body of data on different known normal and ecstatic patient groups, showing that the biomechanical indices successfully confirm the categories of the patients.
The results are of real interest. However, it seems to me that the present analysis (useful though it is) could be interpreted as merely showing that , with patients who are already known to be ectatic, the biomechanical results agree. The analysis does not demonstrate that the biomechanical measurements offer “extra” information which improves, e.g., the ability to detect early ectasis or to monitor changes more closely. It would be useful to include more discussion of this question in the end section of the paper.
While appreciating that some of the figures are simply the outputs from the commercial instrument, I don’t think that they’re always useful to readers in this form, since the scale of reproduction renders the information illegible. I’ve pointed out some of these problems in my comments.
Specific points:
Line 61. Surely rigid contact lenses can give quite good results with some patients?
Table 1: Should “1st applanation” appear immediately below the Table title, rather than above the 1st set of entries in the Table?
Fig.7 Font size in all labels is much too small. Please increase so that labels can be read more easily
Fig.8 I assume that this is the output from the instrument and the format will be familiar to regular users of the device but there’s far too much (partly illegible) information for the average reader. Why not just present the panels giving the information referred to in the text?
Fig.9. Again the font sizes used in this figure are too small to be easily legible. The caption should explain what the two right-hand panels represent (I think that it’s A- C). If so, why isn’t B- D also shown?
Fig.10. I don’t understand why a mass of illegible output from the display is shown here. Why not just present those panels that are of interest and explain clearly what they show? Or give the parameters in a Table?
Line 328. Perhaps it might be better to say that “slit-lamp biomicroscopy confirmed that he had undergone hexagonal incision surgery “
Fig.13. Again, I’m not sure that reproducing this report, with its mass of almost illegible detail will be very useful to the average reader. The only thing that is commented on in the text is the high CBI value (which after much effort I located as being given in the small panel at the bottom right) This figure could be dropped.. In fact, do the biomechanical results really add very much to the interpretation of this patient’s problems?
Author Response
As the authors make clear, biomechanical measurements of the cornea may, in principle, give valuable additional information to supplement other methods for the diagnosis, prognosis and monitoring of ectatic conditions. The present manuscript offers a review of available methods for quantifying biomechanical results in terms of various indices and applies these to the analysis of a large body of data on different known normal and ecstatic patient groups, showing that the biomechanical indices successfully confirm the categories of the patients.
The results are of real interest. However, it seems to me that the present analysis (useful though it is) could be interpreted as merely showing that , with patients who are already known to be ectatic, the biomechanical results agree. The analysis does not demonstrate that the biomechanical measurements offer “extra” information which improves, e.g., the ability to detect early ectasis or to monitor changes more closely. It would be useful to include more discussion of this question in the end section of the paper.
We appreciate the reviewer’s comment. We added a paragraph ate conclusion section – line 265.
While appreciating that some of the figures are simply the outputs from the commercial instrument, I don’t think that they’re always useful to readers in this form, since the scale of reproduction renders the information illegible. I’ve pointed out some of these problems in my comments.
Specific points:
Line 61. Surely rigid contact lenses can give quite good results with some patients?
We believe rigid contact lenses can surely give good results in some cases, but some patients still seek for refractive surgery due to unsatisfactory quality of vision.
Table 1: Should “1st applanation” appear immediately below the Table title, rather than above the 1st set of entries in the Table?
We appreciate the reviewer’s comment and fixed that.
Fig.7 Font size in all labels is much too small. Please increase so that labels can be read more easily
We appreciate the reviewer’s comment and fixed that.
Fig.8 I assume that this is the output from the instrument and the format will be familiar to regular users of the device but there’s far too much (partly illegible) information for the average reader. Why not just present the panels giving the information referred to in the text?
We tried to increase figure resolution and also clarified the information in the text.
Fig.9. Again the font sizes used in this figure are too small to be easily legible. The caption should explain what the two right-hand panels represent (I think that it’s A- C). If so, why isn’t B- D also shown?
We tried to increase figure resolution and also clarified the information in the figure legend. We did not show B – D because this would be the differential map from OS. But we chose to show OD only because this eye presented the highest progression.
Fig.10. I don’t understand why a mass of illegible output from the display is shown here. Why not just present those panels that are of interest and explain clearly what they show? Or give the parameters in a Table?
We followed the reviewer’s suggestion and presented the important information.
Line 328. Perhaps it might be better to say that “slit-lamp biomicroscopy confirmed that he had undergone hexagonal incision surgery “
We followed the reviewer’s suggestion.
Slit-lamp biomicroscopy confirmed that he had undergone hexagonal incision surgery, which was performed thirty years ago for hyperopic and astigmatic correction.
Fig.13. Again, I’m not sure that reproducing this report, with its mass of almost illegible detail will be very useful to the average reader. The only thing that is commented on in the text is the high CBI value (which after much effort I located as being given in the small panel at the bottom right) This figure could be dropped.. In fact, do the biomechanical results really add very much to the interpretation of this patient’s problems?
We followed the reviewer’s suggestion.
Reviewer 2 Report
This study provide corneal biomechanical data which enhances the evaluation of patients with corneal ectasia and meaningfully adds to the multimodal diagnostic armamentarium. Besides, the integration of biomechanical data and corneal tomography with artificial intelligence data augments the sensitivity and specificity for screening and enhancing early diagnosis. In conclusion, corneal biomechanics may be relevant for determining the prognosis and staging the disease.
Author Response
We appreciate the reviewer's comments.
Reviewer 3 Report
In this paper, authors make a review of the role of corneal biomechanics for the clinical evaluation of patients with ectatic corneal diseases.
The paper is well written and structured, and conclusions are supported by the results, however, the main conclusion presented (that TBI is the parameter that best discriminates ectatic conditions) has already been presented and validated in several papers from the same authors.
The review presented in Introduction section, that may be of a certain interest for readers to understand the evolution of the study of corneal biomechanics; and the higher number of eyes used in this case, if compared to other studies, would be its strong points.
A few minor corrections are suggested to improve the quality of the study:
line 95 - IR light - Infrared Light? (Please define the acronym before using)
line 135 - IOP - Intra ocular pressure? (Please define the acronym before using)
line 204-205 - If there were 113 unoperated ectatic eyes from patients with very asymmetric ectasia and 125 fellow eyes with normal topographic maps, this means that 12 ectatic eyes were discarded for the study. What were the reasons?
Discussion section appears after conclusions section, and both are numbered "4"
Author Response
line 95 - IR light - Infrared Light? (Please define the acronym before using)
Answer: We appreciate the reviewer suggestion and changed it in the text.
line 135 - IOP - Intra ocular pressure? (Please define the acronym before using)
Answer: We appreciate the reviewer suggestion and changed it in the text.
line 204-205 - If there were 113 unoperated ectatic eyes from patients with very asymmetric ectasia and 125 fellow eyes with normal topographic maps, this means that 12 ectatic eyes were discarded for the study. What were the reasons?
Answer: We excluded these eyes because they were submitted to some king of surgical procedure, such as stromal ring segment implantation or Atens Protocol.
Discussion section appears after conclusions section, and both are numbered "4"
Answer: We appreciate the reviewer suggestion and changed it in the text.
Round 2
Reviewer 3 Report
Authors have cleared all my concerns